



# Interannual variability of sea level in the South Indian Ocean: Local versus remote forcing mechanisms

Marion Kersalé[1,2], Denis L. Volkov[1,2], Kandaga Pujiana[1,2], Hong Zhang[3]

[1]Cooperative Institute for Marine and Atmospheric Studies, University of Miami, Miami, Florida, USA
[2]NOAA Atlantic Oceanographic and Meteorological Laboratory, Miami, Florida, USA
[3]Jet Propulsion Laboratory, California Institute of Technology, Pasadena, California, USA

*Correspondence to*: Marion Kersalé (marion.kersale@gmail.com)

**Abstract.** The subtropical South Indian Ocean (SIO) has been described as one of the world's largest heat accumulators due to its remarkable warming during the past two decades. However, the relative contributions of the remote (of Pacific origin)
forcing and local wind forcing to the variability of heat content and sea level in the SIO have not been fully attributed. Here, we combine a general circulation model, an analytic linear reduced gravity model, and observations to disentangle the spatial and temporal inputs of each forcing component on interannual to decadal timescales. A sensitivity experiment is conducted with artificially closed Indonesian straits to physically isolate the Indian and Pacific Oceans, thus, intentionally removing the Indonesian throughflow (ITF) influence on the Indian Ocean heat content and sea level variability. We show that the relative
contribution of the signals originating in the equatorial Pacific versus signals caused by local wind forcing to the interannual variability of sea level and heat content in the SIO is dependent on location within the basin (low vs. mid latitude; western vs. eastern side of the basin). The closure of the ITF in the numerical experiment reduces the amplitude of interannual-to-decadal sea level changes compared to the simulation with a realistic ITF. However, the spatial and temporal evolution of sea level patterns in the two simulations remain similar and correlated with El Nino Southern Oscillation (ENSO). This
suggests that these patterns are mostly determined by local wind forcing and oceanic processes, linked to ENSO via the 'atmospheric bridge' effect. We conclude that local wind forcing is an important driver for the interannual changes of sea level, heat content, and meridional transports in the SIO subtropical gyre, while oceanic signals originating in the Pacific amplify locally-forced signals.

## 1 Introduction

The Indian Ocean has been characterized as one of the major heat accumulators in the ocean with a significant increase of ocean heat content (OHC) and sea level during the past two decades, in particular in the Southern Indian Ocean (SIO) (Roemmich et al., 2015; Nieves et al., 2015; Volkov et al., 2017; Zhang et al., 2018; Ummenhofer et al., 2020) (Figure 1a). Heat accumulation in the SIO has resulted in the acceleration of regional sea level rise (Jyoti et al., 2019), more frequent and



intense marine heat waves (Oliver et al., 2018), impacts on Indian summer monsoon rainfall (Venugopal et al., 2018; Gao et
al., 2018), and on increased inter-ocean heat transport via the Agulhas leakage (Wang, 2019). Understanding the
mechanisms of the regional sea level and heat content variability is, therefore, necessary for improving regional weather and
climate forecasts (Levitus et al., 2005; Church and White, 2011; IPCC, 2021).

The decade-long increase of the upper-ocean heat content in the SIO in 2005-2013 (Figure 1a) has been attributed to an
enhanced heat transport from the equatorial western Pacific, carried by the Indonesian throughflow (ITF; Lee et al., 2015).
The ITF is driven by the inter-ocean pressure gradient that exists between the western equatorial Pacific and the eastern
Indian Oceans (Wyrtki, 1987), and serves as an important component of the global thermohaline circulation (Gordon, 1986;
Sloyan and Rintoul, 2001;Sprintall et al., 2014). With a time-mean volume transport of about 15 Sverdrups (1 Sv=$10^6$ m$^3$ s$^{-1}$;
Sprintall et al., 2009), the ITF experiences substantial interannual variations linked to the El Niño/Southern Oscillation
(ENSO; Meyers, 1996; England and Huang, 2005). ENSO is characterized by alternating positive (El Niño) and negative (La
Niña) anomalies in the upper-OHC in the eastern equatorial Pacific, as demonstrated by the Multivariate ENSO Index (MEI
- blue and red shading in Figure 1a). These anomalies eventually impact the SIO by modulating the ITF volume and heat
transport - the so-called "ocean tunnel effect" (Lee et al., 2015; Zhang et al., 2018; Gordon et al., 2019; Volkov et al., 2020).
La Niña conditions favor a stronger ITF, while El Niño conditions are associated with a weaker ITF (Gordon et al., 1999;
Sprintall and Revelard, 2014). The majority of the ITF waters flows westward with the South Equatorial Current (SEC)
between 10°-20°S, while part of the ITF makes its way to the southward-flowing Leeuwin Current (LC) off the West
Australian coast (Thompson, 1984; Domingues et al., 2007). Temperature anomalies carried by the ITF into the SIO can also
propagate along the West Australian coast southward as coastally trapped waves (Clarke, 1991; Clarke and Liu, 1994;
Wijffels and Meyers, 2004).


Besides the "ocean tunnel effect", the variability of sea level and OHC in the SIO is also driven by local atmospheric
forcing. The ocean dynamics due to wind-stress forcing is the dominant contributor of the upper-ocean temperature
variability in the equatorial Indian ocean (Yuang et al., 2020). Wind stress curl induces Ekman pumping, and along-shore
winds cause coastal upwelling and downwelling. These processes alter the depth of the thermocline and, therefore, drive
regional changes in sea level and OHC. Wind forcing over the SIO is related to ENSO due to atmospheric teleconnections
via the Walker Circulation (Schott et al., 2009). The wind-driven changes of sea level and OHC in the Indian Ocean
associated with ENSO are often referred to as the "atmospheric bridge" effect. The "atmospheric bridge" acts mostly through
changes in trade and equatorial winds. Equatorial wind anomalies generate differences in sea surface temperature (SST)
anomalies between the western and eastern equatorial Indian Ocean, termed the Indian Ocean Dipole (IOD; Saji and
Yamagata, 2003) and represented by the Dipole Mode Index (DMI; black line in Figure 1a). The IOD impacts sea level and
heat content in the SIO through changes of the thermocline depth to the west and south of Sumatra and Java, which can also





modulate the ITF transport (Sprintall et al., 2009; Drushka et al., 2010; Lu et al., 2018; Pujiana et al., 2019). Although sometimes ENSO and IOD are regarded as a single entity (Allan et al., 2001), there is more evidence that the two processes are independent from each other (Ashok et al., 2003). Finally, the Southern Annular Mode (SAM) was diagnosed as the
ultimate large-scale climatic forcing over the SIO (Schott et al., 2009). The SAM has significant effects on the supergyre, which links all the three subtropical gyres of the Southern Hemisphere (Cai et al., 2005; Ridgway and Dunn, 2007). SAM further contributes to modulating the teleconnections between ENSO and IOD by enhancing sea surface temperature (SST) gradients within the SIO (Cleverly et al., 2016).

The signals of the Pacific origin that enter the SIO with the ITF and reach the West Australia coast radiate westward as eddies and Rossby waves (Cai et al., 2005, Zhuang et al., 2013; Zheng et al., 2018). This process is called hereafter remote or eastern boundary forcing. On seasonal to interannual time scales, local wind forcing and related Ekman pumping over the ocean interior can also generate oceanic Rossby waves and/or alter those emanating from the eastern boundary (Masumoto and Meyers, 1998; Birol and Morrow, 2001). Menezes and Vianna (2019) found that the westward propagation observed in
the Eastern SIO (ESIO - Figure 1b) basin and in the Western SIO (WSIO - Figure 1b) basin reveals a superposition of Rossby waves generated by processes near the eastern boundary on top of Rossby waves generated by Ekman pumping in the mid-basin. Thus, westward propagation provides the primary mechanism for both the "ocean tunnel" and the "atmospheric bridge" effects to transfer energy and seawater properties across the entire SIO (Morrow and Birol, 1998)

On the basis of satellite altimetry, previous studies have examined the relative importance of the eastern boundary and the local wind forcing on the interannual variability of sea level and heat content in the SIO. Volkov et al. (2020) showed that the signals of the Pacific origin dominate sea level variability in the ESIO, while the local wind-induced variability is more influential in the WSIO, indicating a dependence on longitude. They also noted that the relative importance of local and remote drivers in the WSIO is time-dependent: in some periods remote and local forcing mechanisms appear to have similar
magnitudes, while in other periods local wind forcing may become dominant. The latter happened in 2016-2018 when the OHC and sea level in the SIO recovered after an abrupt drop associated with the 2014-2016 El Niño (Figure 1a). Nagura and McPhaden (2021) extended the analysis of the eastern boundary and local wind forcing mechanisms by also demonstrating a latitudinal dependence. They confirmed that the influence of the eastern boundary forcing is confined to the ESIO but only at low latitudes (from 10° and 17°S). Between 10°S and 35°S, they identified two regimes: one in the tropics where local
forcing has a bigger effect, and another in the mid latitudes (subtropics) where remote forcing has a greater influence. These results are in agreement with earlier studies that suggested the dominance of local wind forcing in driving sea level variability at low latitudes (from 11° to 13°S - Masumoto and Meyers, 1998; Zhuang et al., 2013), and the prevalence of variability radiated from the eastern boundary at mid latitudes (from 20° to 25°S - Zhuang et al., 2013; Menezes and Vianna, 2019).






It is usually assumed that the interannual variability of OHC and sea level along the West Australia coast is strongly linked to remote wind forcing in the tropical Pacific and, therefore, represents the "ocean tunnel" effect (*e.g.*, Nagura and McPhaden, 2021, and references therein). However, the impact of the along-shore wind forcing that also drives sea level variability along the coast is often disregarded. The along-shore winds are part of the large-scale atmospheric circulation
over the SIO dominated by southeasterly trades, the variability of which is modulated by ENSO via the "atmospheric bridge" effect.

The objective of this study is to revisit the question of the interplay between remote and local drivers causing interannual-to-decadal sea level variability in the SIO with a different approach. Specifically, we perform an ocean model sensitivity
experiment, in which we physically isolate the Indian and Pacific Oceans by closing the Indonesian straits, thus removing the ITF ("ocean tunnel") influence on the Indian OHC and sea level variability. In this experiment, any variability observed along the SIO eastern boundary is primarily due to local forcing by construct. The solutions of this experiment were investigated by comparing them to a simulation with open Indonesian passages. By comparing these two runs, we can better quantify the relative contribution of the remote vs local drivers. In addition, numerical simulations are combined with a
linear reduced gravity model and observations to disentangle the spatial and temporal dominance of each forcing component. Furthermore, we investigate the recent significant changes in heat content and associated sea level during two specific time periods associated with the accumulation of the upper 2000-m heat content in the SIO subtropical gyre in 2004-2013 and the abrupt cooling in 2014-2016 (Figure 1a). We finally investigate how the interior meridional transports across the SIO are related to sea level variability.

**2 Materials and Methods**

**2.1 Ocean Model**

The ocean model used in this study is a global ocean and sea-ice state estimate based on the Massachusetts Institute of Technology General Circulation Model (MITgcm, Marshall et al., 1998) and produced by Estimating the Circulation and Climate of the Ocean (ECCO) consortium (https://ecco-group.org). The ECCO consortium aims to create accurate,
physically consistent, time-evolving estimates of ocean circulation by combining MITgcm with selected in situ and satellite observations (Menemenlis et al., 2005; Wunsch and Heimbach, 2007; 2013; Forget et al., 2015; Fukumori et al., 2017). The estimate uses the adjoint method to iteratively minimize the squared sum of weighted model-data misfits and to adjust the model control parameters (Wunsch et al., 2009; Wunsch and Heimbach, 2013). The data constraints include temperature and salinity profiles (from Argo floats, CTD, XBT, and ice-tethered profilers), satellite altimetry and gravimetry measurements,
sea surface temperature fields from passive microwave radiometry, and satellite observations of sea-ice concentration. The control parameters include the initial temperature and salinity fields, the 3D parameters of Gent-McWilliams/Redi mixing scheme (Redi, 1982; Gent and McWilliams, 1990), and the time-varying atmospheric boundary conditions.



Presently, all ECCO models employ a so-called Lat-Long-Cap (LLC) grid, ranging from LLC90 (~1°), LLC270 (~1/3°), to
LLC4320 (~1/48°), which allows for an improved representation of the Arctic (no polar singularity and fine grid for small
deformation radius). The LLC grid has five faces covering the whole globe, with a simple, locally isotropic latitude-
longitude grid between 70°S and 57°N and an Arctic cap (Forget et al., 2015). In this study, we use the ECCO LLC270
configuration (Zhang et al., 2018), which provides a better representation of mesoscale variability compared to the latest
ECCO LLC90 release (ECCO-V4r4, Forget et al., 2015). The horizontal resolution of the LLC270 grid varies spatially from
12 km at high latitudes to 28 km at midlatitudes. The vertical grid has 50 vertical layers, with the spacing increasing from 10
m near the surface to 457 m near the maximum model depth set to 6145 m.

The ECCO LLC270 solution used in this study is obtained by a free forward model integration from January 1992 to June
2018 using the adjusted control parameters and forced by the adjusted ERA-Interim atmospheric fields (Dee et al., 2011). To
separate the influence of the Pacific Ocean on the variability of sea level and OHC in the Indian Ocean, we performed a
numerical experiment with artificially closed Indonesian passages and the Torres Strait between Australia and New Guinea.
The closure of the Indonesian passages is achieved by modifying the model bathymetry (Figure 2a). To distinguish between
the optimized, realistic model run with open Indonesian passages and the experiment with closed Indonesian passages, we
refer to them hereafter as the ITF-on and the ITF-off experiments, respectively. The ITF-off experiment was run twice. The
first run was performed to let the model reach a stable state. The second run was initialized using the output of the first run in
January 2018, and it was integrated again from January 1992 to June 2018. From now on, the ITF-off experiment refers to
the second ITF-off run only. Because the ECCO LLC270 is a volume-conserving Boussinesq model, it does not reproduce
the global mean sea level change. Therefore, the global mean sea level is subtracted prior to the analysis (Greatbatch, 1994).

### 2.2 Model for sea level variability

The westward propagation of Rossby waves and eddies provide the primary mechanism for both the "ocean tunnel" and the
"atmospheric bridge" effects to transfer energy across the SIO. Under the long-wave approximation, these processes can be
quantified by a 1.5-layer reduced-gravity model (e.g., Qiu, 2002), which has also been widely used to investigate the
variability of sea level in the SIO (e.g., Zhuang et al., 2013; Jin et al., 2018; Menezes and Vianna, 2019; Volkov et al., 2020;
Nagura and McPhaden, 2021). This reduced-gravity model, hereafter called RG model, is governed by the following linear
vorticity equation, which separates the impacts of sea level signals originating at the eastern boundary and those generated
by local wind forcing:

$$\frac{\partial \eta}{\partial t} = c_R \frac{\partial \eta}{\partial x} - \frac{g' \nabla \times \tau'}{\rho_0 g f} - \epsilon \eta \qquad (1),$$





where $\eta$ is the baroclinic component of sea level anomaly (SLA), $x$ is the longitude, $t$ is the time, $c_R$ is the zonal phase speed of long baroclinic Rossby waves, $g$ is the gravitational constant, $g'$ is the reduced gravity, $\tau'$ is the wind stress anomaly vector, $\rho_0$ is the mean sea water density, $f$ is the Coriolis parameter, and $\varepsilon$ is the damping coefficient.

Integrating Eq. (1) westward from the eastern boundary ($x_e$) along the baroclinic Rossby wave characteristics yields the following solution (Qiu, 2002):

$$\eta(x,t)=\eta\left(x_e,t-\frac{x-x_e}{c_R}\right)e^{\frac{-\epsilon}{c_R}(x-x_e)} - \frac{g'}{\rho_0 gf c_r}\int_{x_e}^{x} \nabla \times \tau'\left(x',t-\frac{x-x'}{c_R}\right)e^{\frac{-\epsilon}{c_R}(x-x')} dx' \qquad (2).$$

The first term on the right side of Eq. (2) represents the SLA signal that originates at the eastern boundary. The second term represents the SLA signal generated by the local wind stress curl. Both signals propagate westward and decay at a rate determined by the damping coefficient, $\varepsilon$. The RG model was initialized with the low-pass-filtered SLA at 110°E, $\eta(x_e, t)$, obtained from the two numerical simulations (ITF-on and ITF-off) at 13°S and 25°S (similar to Nagura and McPhaden, 2021). The model was forced with the ECCO LLC270 wind stress and integrated from 110°E to 50°E. It should be noted that the eastern boundary at 13°S is located further east at ~130°E. However, because $\eta(x_e, t)$ at 110°E and 130°E are strongly correlated (R=0.80), we initialize the model for both latitudes at 110°E for consistency. The RG model parameters were obtained empirically, and they are summarized in Table 1. Specifically, the phase speeds ($c_R$) were set to 6.5 cm s$^{-1}$ at 25°S and 13 cm s$^{-1}$ at 13°S, based on the Hovmöller diagrams of the altimetry and model SLA in the SIO (see Section 3.1). These phase speeds are consistent with those estimated by Nagura and McPhaden (2021). The optimal values of g' and ε were obtained iteratively using a linear regression. The zonal mean g' varies from 0.05 m s$^{-2}$ at 25°S to 0.07 m s$^{-2}$ at 13°S for the ITF-on experiment and from 0.04 m s$^{-2}$ at 25°S to 0.07 m s$^{-2}$ at 13°S for the ITF-off experiment, reflecting the meridional variation in stratification (Zhuang and al., 2013). A regression analysis yielded the following optimum damping coefficients: $\varepsilon^{-1}$=2.5 years at 13°S and $\varepsilon^{-1}$=1.9 years at 25°S for the ITF-on experiment, and $\varepsilon^{-1}$=3.3 years at 13°S and $\varepsilon^{-1}$=1.2 years at 25°S for the ITF-off experiment (Table 1).

**2.3 Data**

Steric or density-driven changes dominate large-scale sea level variability on seasonal-to-interannual time scales (e.g., Gill and Niller, 1976). The monthly maps of altimetry SLA from 1993 to 2019 provided by the Copernicus Marine Environment Monitoring Service (CMEMS; Ducet et al., 2000) are used to validate the sea level changes simulated by the ECCO LLC270 model. The global mean sea level rise was subtracted from SLA maps to focus on regional variations. Since density variations in low and mid-latitudes are mainly caused by temperature variations, satellite measurements of sea level can be used as a proxy for OHC (Roemmich and Gilson, 2006). To illustrate this point in the SIO, we compared the OHC anomaly time series for the upper 2000 m (Levitus et al., 2012) with the altimetry SLA variability averaged over 55°E to 115°E and





10°S to 30°S (green and red lines in Figure 1a, respectively). The time series are strongly correlated (R=0.90) confirming that SLA is a good indicator of OHC changes in the SIO.


We use the MEI, produced by the National Oceanic and Atmospheric Administration (NOAA) Earth System Research Laboratory's Physical Sciences Division (www.esrl.noaa.gov/psd), which incorporates both oceanic and atmospheric variables to provide a single index of ENSO intensity. This monthly index integrates the impact of five factors over the tropical Pacific basin (sea level pressure, sea surface temperature, zonal and meridional components of the surface wind, and

outgoing long-wave radiation). Positive MEI events are related to warm, El Niño conditions and negative MEI events to cold, La Niña conditions (Figure 1a). As indicators of IOD, we employed the DMI, a monthly product of the NOAA Earth System Research Laboratory (Saji and Yamagata, 2003). DMI is defined as the difference between averaged sea surface temperature over the western equatorial Indian Ocean (50°E-70°E; 10°S-10°N) and the south eastern equatorial Indian Ocean (90°E-110°E; 10°S-0°). Finally, we used the monthly Marshall SAM index based on available station zonal pressure

observations between 40°S and 65°S (Marshall, 2003) obtained from British Antarctic Survey's website (www.nerc-bas.ac.uk/icd/gjma/sam.html).

## 2.4 Statistical analysis

To identify the dominant spatio-temporal modes of SLA variability in the Indian Ocean, an Empirical Orthogonal Function (EOF) analysis was carried out using the Matlab Climate Data Toolbox (Monahan et al., 2009; Greene et al., 2019). The

obtained spatial patterns of the variability are referred to as EOFs, and their temporal evolutions are shown by the Principal Component time series (PCs). Prior to computing the EOFs, the global mean sea level, the seasonal cycle, and the linear trend were subtracted from the data. Then the data were low-pass filtered with a cutoff period of 1 year to focus on the interannual-to-decadal variability. The spatial patterns of EOFs are represented as regression maps obtained by projecting SLA onto the standardized (divided by standard deviation) PC time series. Thus, the regression coefficients are in

centimeters (local change of sea level) per 1 standard deviation change of the PC. The linear regression analysis was also performed to relate the SLA to atmospheric circulation patterns (i.e., the meridional surface wind stress). Using these regression coefficients, we can reconstruct a SLA time series at each grid point associated with the PC or the meridional wind stress variability.

## 3 Results

### 3.1 Variability of sea level in the Indian Ocean

Before exploring the ocean processes related to sea level changes in the ITF-on and ITF-off experiments, we first validate the model's performance by comparing the modeled upper-ocean ITF transport with that obtained from moored velocity





measurements across the Makassar Strait, which is the primary ITF gateway (Figure 2a; Gordon et al., 1999, 2010; Pujiana et al., 2019). The agreement between the two monthly transport time series is rather good (Figure 2b), with a correlation of 0.81. Over the period of 2004-2017 with a gap from October 2008 to August 2009, the observed and simulated mean ITF transports are -12.5 Sv and -11.6 Sv (minus sign indicates southward transport), respectively. The low-pass filtered time series clearly shows a pronounced reduction of the upper-ocean ITF transport in 2014-2016 in both the ITF-on experiment and in observations. This anomalous ITF transport is primarily a response to the strongest El Niño to date of the 21st century followed by the strongest on record negative IOD (Figure 1a; Pujiana et al., 2019). The record maximum transport in 2017 reported by Gordon et al. (2019) is also well simulated by the model. The realistic representation of the ITF in the ECCO LLC270 model gives confidence that closing the Indonesian passages in the ITF-off experiment will help to better understand the impact of the ITF on the variability of sea level and OHC in the SIO. The closure of the ITF leads to a weakening of the SEC and the LC, which is demonstrated by the difference between the time-mean upper-100 m velocities in the ITF-off and the ITF-on experiments (Fig. 2a). This is consistent with the results of Lee et al. (2002), who also conducted a numerical experiment with closed Indonesian passages.

The 2004-2013 linear trend of SLA derived from satellite altimetry (Figure 1b) shows two well-outlined regions of positive trend values: the first region occupies most parts of the Northern and Equatorial Indian Ocean, while the second region is confined to the SIO extending off the western coast of Australia westward between 55°E to 115°E and 10°S to 30°S (Box A). The time series of altimetry SLA averaged over Box A shows a persistent increase in 2004-2013 (Figure 3a). In 2014, sea level reached a decadal maximum, after which it decreased abruptly following the 2014-2016 El Niño. After reaching a local minimum in 2016, it partially recovered by the end of 2017. The observed SLA variability is well simulated by the ITF-on experiment (compare red and black curves in Figure 3a; R=0.80). In contrast, there was no pronounced increase of SLA in 2004-2013 in the ITF-off experiment (black dashed curve in Figure 3a), which suggests the dominant role of the ITF in driving this decade-long change. The ITF-off experiment also shows a decrease of SLA in 2014-2016, although the magnitude of this decrease was smaller than in the ITF-on simulation by about a factor of two (solid and dashed black curves in Figure 3a; R=0.38 between the two simulated SLA).

It has been shown that the relative importance of remote and local drivers for sea level variability in the SIO depends on longitude and latitude (Volkov et al., 2020; Nagura and McPhaden, 2021). Based on these findings, we examined the SLA averaged over three different areas within box A (see insert in Figure 3). The time series of the observed and simulated SLA averaged over the WSIO at low latitudes (from 10° and 17°S; box B in Figure 3) are closely aligned throughout the entire period (R=0.94). It appears that the closure of the Indonesian passages does not significantly change the interannual variability of SLA in this area (Figure 3b; R=0.82 between the observed and simulated SLA). Decadal tendencies are also similar in the two simulations over the ESIO at low latitudes (from 10° and 17°S; box C), although the interannual variability is somewhat different (Figure 3c; R=0.78 in the ITF-on experiment and R=0.14 in the ITF-off experiment). In contrast, the





time series of SLA averaged over the SIO at mid latitudes (from 17° and 35°S; box D) exhibit behaviors similar to the SLA time series averaged over the entire box A: (i) there is no significant increase of SLA in 2004-2013 and (ii) the magnitude of the 2014-2016 SLA is reduced by a factor of three in the ITF-off simulation compared to the ITF-on simulation (Figure 3d).

The correlation coefficients between the SLA time series observed by satellite altimetry (red curves) and simulated in the ITF-on (black solid curve) and the ITF-off (black dashed curve) experiments are 0.86 and 0.45, respectively. These results suggest that the decade-long increase of SLA in 2004-2013 and the development of SLA anomaly in the SIO in 2014-2016 are mostly due to remote forcing at mid-latitudes, while local forcing is more important at low-latitudes.

While investigating the relative importance of each mode of atmospheric variability goes beyond the scope of this paper, it has been shown that all these climate modes can influence the interannual-to-decadal variability of sea level in the SIO but none of them alone is able to explain the whole complexity of the local wind forcing (Volkov et al., 2020). The squared correlation coefficients between the ENSO, IOD and SAM indices (see Section 2.3) and the SLA averaged over the boxes B, C and D quantify to what extent the variance of these three climate modes explain the variance of SLA (percent variance).

ENSO explains 58% of the SLA variance in box B with no time-lag, while its influence is significant in box C with a time-lag of 13 months (percent variance of 36% ). No other significant relationships were found.

The westward propagation of interannual SLA signals has been the most prevalent characteristics of the spatio-temporal SLA variability in the SIO over the entire period of both the altimetry observations and the numerical simulations, as seen in

the Hovmöller diagrams of SLA (Figure 4a-c). The altimetry data (Figure 4a) and the ITF-on simulation (Figure 4b) show similar signals detected at the eastern boundary reaching the western boundary 2-3 years later. For example, the two strongest positive SLA at the eastern boundary in 1999-2001 and in 2010-2013 associated with La Niña conditions reached 50°E in 2002-2003 and in 2014, respectively. A negative SLA driven by the 1997-1998 El Niño reached 50°E in 2000. It is interesting to note that the amplitude of the 1997-1998 (1999-2001) anomaly increased (decreased) to the west of the Ninety

East Ridge. It also appears that local processes can sometimes suppress and prevent signals from crossing the Ninety East Ridge. The 2003–2007 negative anomaly is an example of a negative SLA signal that did not travel far past the Ninety East Ridge. The signals simulated in the ITF-off experiment (Figure 4c), including those emanated from the eastern boundary (Figure 4d), generally have smaller amplitudes than those observed in the ITF-on simulation and in observations, but the spatial and temporal structures of the signals are similar. This indicates that local processes and wind forcing in the SIO,

possibly related to the atmospheric bridge effect, are important drivers for the regional sea level and OHC changes.

The large-scale spatio-temporal changes of SLA were also studied with an EOF analysis. In the ITF-on simulation, the leading EOF (EOF1; Figure 5a) explains 44% of the interannual SLA variance, and it reveals a dipole structure with a positive anomaly over the central and western tropical Indian Ocean and a negative along the eastern boundary and around

the Maritime continent (Figure 5a). The regression of wind stress on the PC1 (Figure 5a) shows that the first EOF is linked





to ENSO associated with weaker trade winds in the SIO and easterly wind anomalies along the equator. This atmospheric circulation pattern favors upper-ocean warming in the northwestern SIO and cooling in the southeastern SIO, leading to positive and negative SLA, respectively. The EOF1 in the ITF-off simulation explains 41.1% of the variance (Figure 5b), and it displays very similar large-scale features. However, one can notice that the positive loading of the dipole in the central Indian Ocean is intensified and extends further east and the area of cold anomaly adjacent to the western Australian coast becomes narrower. It can probably be explained by the weaker SEC and LC in the ITF-off experiment (Figure 2a). This comparison suggests that the spatial variability pattern is mostly determined by local processes not related to the "ocean tunnel" effect.

The PC1 time series of both the ITF-on and the ITF-off simulations are highly correlated (R=0.97; Figure 5c,d). There are two distinct peaks in 1997-1998 and in 2014-2016 that coincided with the two strongest on record El Niño events (Figure 1a). We reconstructed SLA using only the EOF1 for both the ITF-on and the ITF-off experiments. The local SLA variance explained by the EOF1 is shown in Figure 5 (e and f). One can see that the reconstruction explains greater than 70% of the SLA variance over the central and western tropical Indian Ocean and along the eastern boundary in both experiments. While the experiments display very similar large-scale patterns, the EOF1 for the ITF-on simulation is responsible for somewhat greater variance explained along the eastern boundary.

Along with the signals coming from the equatorial Pacific, the variability of sea level and OHC along the West Australia coast is also driven by along-shore winds. These winds are modulated by ENSO via the "atmospheric bridge" effect. The regression of the monthly modeled SLA on the monthly wind stress averaged over 110-115°E and 20-35°S (rectangles in Fig. 6 a,b) is presented for both the ITF-on (Fig. 6 a,c) and ITF-off (Fig. 6 b,d) simulations. The regression shows that northward/southward wind anomalies along the West Australia coast favor lower/higher sea level along the coast, with a lower amplitude in the ITF-off experiment compared to the ITF-on simulation. As expected, the results suggest the importance of upwelling/downwelling off the coast favored by southerly/northerly wind anomalies. Note that the area close to the West Australia coast is characterized by the strongest positive meridional wind stress in the Indian Ocean and elevated standard deviations (not shown). The reconstruction of SLA using the obtained regression coefficients demonstrates that the along-shore winds over the 1992-2018 time period explain 15% and 11% of the SLA variance in the ESIO in the ITF-on and ITF-off simulations, respectively. It should be noted that this relationship is time dependent, and becomes significant if we focus on the 2004-2018 time period. Over this time period, the along-shore winds explain 25% and 36% of the SLA variance in the ESIO in the ITF-on and ITF-off simulations, respectively.





### 3.2 Local versus remote forcing mechanisms

We quantify and assess here the combined and relative contributions of the eastern boundary forcing and the local wind stress curl to the interannual variability of the SIO using the 1.5-layer RG model (see Section 2.2) at 13°S and 25°S for both the ITF-on and the ITF-off simulations.


In the ITF-on experiment, the RG model reproduces SLA reasonably well (Figure 7a,b). The local wind forcing appears to dominate over the eastern boundary forcing at 13°S in the WSIO (Figure 7b,d). The surface forcing from the local wind stress curl (Figure 7d) exhibits SLA of similar amplitude with negative anomalies in 2000-2003, positive in 2004-2008 and negative in 2008-2015. The boundary driven part shows mainly positive anomalies in 1999-2003 and negative anomalies in
2003-2008 (Figure 7c). After 2008, the SLA from the eastern boundary shows alternate positive and negative values. The Hovmöller diagrams of SLA at 25°S indicate that the total solution compares well with both the eastern boundary forcing and the local wind forcing. In the ESIO at this latitude, the boundary driven part of the RG model solution has a higher amplitude and so a higher impact than the wind driven part (Figure 7g,h).

The relative contribution of eastern boundary and local wind stress forcing terms is estimated by computing the explained variance between the ECCO and the RG modeled SLA. Over the 2004-2013 time period, the SLA variability in the WSIO is mostly driven by the local wind forcing at 13°S (red line in Figure 8a; 58% of SLA change), while the eastern boundary forcing has a stronger relative impact at 25°S in the WSIO (blue line in Figure 8c; 9% of SLA change). The eastern boundary forcing is dominant at both latitudes in the ESIO during this time period (blue lines in Figure 8b,d; between 30 and
86% of SLA change). Between 2014-2016, one can see an increasing importance of the local wind forcing in the ESIO at 13°S (22% of SLA change) and of both the boundary forcing (61% of SLA change) and the wind forcing (76% of SLA change) in the WSIO at 25°S.

We also applied the 1.5-layer RG model at 13°S and 25°S for the ITF-off simulation (Figure 9). Once again, the RG model is
able to reproduce the simulated ECCO LLC270 SLA (Figure 9a,b). The signals simulated in the ITF-off experiment (Figure 9) generally have the same characteristics as those simulated in the ITF-on experiment but with smaller amplitudes at 25°S (Figure 7). The local wind forcing appears to be the main driver for SLA variability at 13°S in the WSIO (Figure 9b,d), and the eastern boundary forcing dominates at 25°S in the ESIO (Figure 9f,g).

Closing the Indonesian passages in the ITF-off experiment increases the importance of wind forcing over the WSIO at both latitudes in 2004-2013 (Figure 10a,c; 56-87% of SLA change). The sea level variability during this time period in the ESIO is still mainly attributed to the eastern boundary forcing (Figure 10b,d; ~42-78% of SLA change). During the 2014-2016



time period, the sea level variability is induced by both the local wind forcing (~64-95% of SLA change) and the eastern boundary forcing (~29-83% of SLA change) in the four subdomains of the SIO.

## 3.3 Meridional transport of the subtropical gyre

The large-scale sea level variability in the subtropical SIO reflects the meridional transports associated with the Meridional Overturning Circulation in the Indian Ocean (Zhuang et al., 2013; Nagura, 2020). Due to the connection between the Pacific and Indian Oceans via the Indonesian passages, these transports play a fundamental role in the inter-ocean exchange of mass, heat, salt, and carbon within the global climate system. A diagnosis of the Indian Ocean meridional transports is thus important for our understanding of the global climate and its variability (e.g., Wang, 2019). The zonal gradient of SLA between the western and eastern regions of the subtropical SIO is a proxy for this zonally integrated meridional transport (Zhuang et al., 2013; Nagura, 2020). Displayed in Figure 11 (c, d) are the zonal differences between SLA averaged over 50°-55°E (SLA$_{west}$) and 110°-115°E (SLA$_{east}$) as a function of latitude (15-30°S) and time in both the ITF-on and the ITF-off simulations. The differences between these two simulations are used to diagnose the relative importance of the ocean tunnel and atmospheric bridge effects on the interannual variability of the meridional transport.

In the ITF-on simulation (Figure 11c), the zonal (west-east) differences of SLA are clearly related to ENSO (see MEI in Figure 11a,b): the positive differences (in 1997-1998, 2002-2007, 2009-2010 and 2014-2016) are generally associated with El Niño conditions (cold anomalies in the ESIO) and the negative differences (in 1999-2001, 2008, 2011-2013) are associated with La Niña conditions (warm anomalies in the ESIO). The correlation between the MEI and the west-east SLA differences in the ITF-on experiment across the latitudinal band considered (15-30°S) ranges between 0.51-0.81, with a maximum correlation at 27°S (not shown). In the ITF-off experiment, the correlation between the west-east SLA differences reduces to 0-0.59, with a maximum correlation at 29°S. Nevertheless, in both the ITF-on and the ITF-off experiments, the SLA gradients exhibit similar amplitudes and patterns. For example, the persistent negative SLA gradients in 1999-2002 and the persistent positive SLA gradients in 2002-2007 are observed in both experiments. However, the shorter-term positive SLA gradients associated with the 1997-1998 and 2014-2016 El Nino events have significantly smaller amplitudes in the ITF-off experiment compared to the ITF-on experiment.

To verify the relationship between the zonal gradient of SLA and the meridional transport in the SIO subtropical gyre, we computed the zonally-integrated transports across 15°S in both the ITF-on and the ITF-off simulations. This latitude is a key position where the meridional transports are used to measure the subtropical cell variability (Zhuang et al., 2013; Nagura, 2020). The transports are integrated from coast to coast thus including contributions from the interior and boundary currents. The transport time series (not shown) at 15°S in both simulations are modestly correlated with the zonal gradients of modeled SLA at 15°S (R=0.49 in the ITF-on experiment and R=0.25 in the ITF-off experiment). The maximum agreement





between the transport time series at 15°S and the zonal gradients of modeled SLA is observed at 25°S, with a correlation of
0.76 in the ITF-on experiment and 0.68 in the ITF-off experiment.

The resulting time-mean streamfunction of the meridional flows is illustrated by the mean profiles of cumulative transport as
a function of depth (Figure 11e,f; black lines). The Indian Ocean is essentially closed in the north aside from the ITF
contribution. Given that, mass conservation requires that the transport integrated over the full depth of the ocean across 15°S
is equal to the ITF transport plus the freshwater balance (precipitation minus evaporation plus river runoff) in the ITF-on
simulation and to the freshwater balance in the ITF-off simulation. The vertically integrated meridional flow yields a time-
mean southward transport of 12.7 Sv in the ITF-on simulation and 0.17 Sv in the ITF-off simulation. The time-mean
cumulative transport in the ITF-on simulation is consistent with the simulated mean ITF transports of -11.6 Sv through the
Makassar Strait, which is the primary ITF gateway (Figure 2b). In the ITF-off simulation, the mean southward flow across
15°S is compensated by the freshwater balance to the north of this latitude (net evaporation of 0.08 Sv estimated between 8°-
20°S in the Indian Basin; e.g. Talley, 2008).

Merging together the positive MEI values for El Niño and negative MEI values for La Niña, we computed the composites of
the obtained streamfunctions associated with the cold anomalies (positive MEI; El Niño; blue lines) and warm anomalies
(negative MEI; La Niña; red lines) in the ESIO in the two experiments (Figure 11e,f). These cold/warm anomalies in the
ESIO favor northward/southward transport anomalies, which is consistent with the results of both experiments. The transport
is amplified/reduced by ± 1 Sv from 300 m to the bottom in the ITF-on experiment. In the ITF-off experiment, the transport
anomaly is observed mainly between 500 and 3000 m with a maximal value of 0.7 Sv. The consistency between the stream
functions demonstrates that the meridional geostrophic transport is modulated by ENSO via the "atmospheric bridge" effect.

## 4. Discussion and Concluding remarks

In this study, we revisit the interplay between the remote and local drivers for the OHC and sea level variability in the SIO
using an eddy-permitting ECCO LLC270 solution. The data-model comparisons demonstrate that the model performs well in
simulating the circulation of the Indian Ocean including the ITF transport. In order to better quantify the relative contribution
of the remote vs local drivers, we conducted a sensitivity experiment with closed Indonesian and Torres straits to physically
separate the Indian and Pacific Oceans and eliminate the ocean tunnel effect on sea level variability in the SIO (ITF-off
experiment).

Effects of the ITF on the circulation and thermal structure of the Indian Ocean were investigated by comparing solutions of
an ocean general circulation model with open and closed Indonesian passages in previous studies (e.g., Hughes et al., 1992;
Lee et al., 2002; Song and Gordon, 2004). These studies focused on seasonal to inter-annual variability and showed that the

ITF warms the upper Indian Ocean and deepens its thermocline. Subsequently, the full role of ITF was analyzed with coupled general circulation models (e.g., Wajsowicz and Schneider, 2001; Song et al., 2007). The changes were similar to those reported by ocean-only model experiments but with larger amplitude. In addition, they showed that the closure of the
ITF significantly increased interannual variability in the eastern tropical Indian Ocean. Closing the ITF shoals the eastern tropical Indian Ocean thermocline, resulting in greater cooling episodes due to enhanced atmosphere-thermocline coupled feedback.

The observed SLA variability in the SIO is well simulated by the ITF-on experiment showing a persistent increase from
2004 to 2014, while no pronounced increase of SLA is observed in the ITF-off experiment (Figure 3a). This result suggests that the observed decade-long heat accumulation is due to the ocean tunnel effect, linked to the ENSO variability. Shutting the ITF off removes heat supply from the Pacific, slows down the SEC, and raises the thermocline in the SIO (Lee et al., 2002). The fact that the observed heat accumulated affected nearly the entire SIO is somewhat inconsistent with numerical tracer experiments showing that the ITF waters entering the Indian Ocean are carried mainly by the SEC (e.g., Song et al.,
2004). Nevertheless, Valsala and Ikeda (2007) showed that the ITF waters can also flow along the west coast of Australia, via LC, and then spread westward across the SIO.

As evidenced by satellite altimetry and Argo measurements, the decade-long accumulation of heat content in the SIO subtropical gyre ended with a strong cooling  in 2014-2016. An abrupt decrease of SLA from 2014 to 2016 was well
simulated in the ITF-on simulation, with an amplitude similar to satellite observations. This time period was marked by a strong El Niño, the associated reduction of southeasterly trade winds and a cyclonic wind anomaly across  the SIO (Volkov et al., 2020). This resulted in Ekman divergence favoring the observed upper-ocean heat content decrease. In addition, the 2014-2016 El Niño was in phase with a weak positive IOD in 2015 and the lowest on record negative IOD in 2016. These phenomena, along with a deepening of the thermocline and increased sea levels along the Maritime continent, resulted in the
reduction of the ITF transport and upper-ocean heat content (Figures 1a, 2b). In 2014-2016, the ITF-off experiment also revealed a decrease in SLA, albeit the magnitude of this decline was two times less than in the ITF-on simulation. This sensitivity experiment confirms that neither the atmospheric bridge nor the ocean tunnel effect alone can fully explain the observed basin-wide cooling.

The EOF analysis revealed that the spatio-temporal structures of the SLA variability in the ITF-on and ITF-off experiments are similar. The difference between the two simulations is mainly limited to the magnitude of the signals. This suggests  that the spatio-temporal structure of the regional SLA and OHC variability is determined by local processes and wind forcing, not related to the ocean tunnel effect.





Winds along the West Australia coast are part of the large-scale atmospheric circulation over the SIO dominated by southeasterly trades and modulated by ENSO via the "atmospheric bridge" effect. We find that these winds are able to explain only 15% and 11% of the SLA variability averaged over 110-115°E and 20-35°S in 1995-2018 in the ITF-on and ITF-off simulations, respectively. The rather small impact of the along-shore wind stress on the regional SLA variability is in agreement with the study of Nagura and McPhaden (2021), who showed that the lead-lag correlation between SLA and wind

forcing does not exceed the 95% confidence level at any lag within ±12-month window. This relationship becomes stronger and significant between 2004 and 2018, where the along-shore winds explain greater SLA variance in the ESIO in the ITF-off simulation compared to the ITF-on simulation. This result shows the importance of the along-shore wind forcing that also drives sea level variability along the coast via the "atmospheric bridge" effect.

The processes driving the interannual variability of sea level and OHC in the SIO were further analyzed in the context of westward-propagating Rossby waves. Confirming previous observation-based studies (Volkov et al., 2020; Nagura and McPhaden, 2021), we showed that the relative importance of signals propagating from the eastern boundary and the local wind forcing varies with latitude (low vs. mid latitude) and longitude (WSIO vs. ESIO). The time series of the simulated SLA averaged over the WSIO at low latitudes (13°S) are closely aligned in both numerical experiments. It appears that the

closure of the Indonesian passages does not significantly change the interannual variability of SLA in this area. Our analysis further supports the idea that the local wind forcing appears to be the main driver for SLA variability in the tropical WSIO. Decadal tendencies are also similar in the two simulations over the ESIO at low latitudes, although the interannual variability is somewhat different. Overall, the results showed that SLA variability in the ESIO is dominated by signals radiated from the eastern boundary. At low latitudes, the interannual sea-level variability is driven by the local wind forcing in the WSIO and

by the eastern boundary forcing in the ESIO. This highlights the longitudinal dependence causing interannual SLA variability in the south Indian Ocean demonstrated by Volkov et al. (2020). The variability of SLA is quite different in the two simulations at mid latitude, although the westward propagation can be easily traced in many cases. Results showed that the interannual variability of SLA at mid latitudes is mainly driven by waves radiated from the eastern boundary related to the ocean tunnel effect, i.e., the ITF and coastally trapped waves that rapidly transfer anomalies generated in the Pacific

along the west Australian coast. This highlights the latitudinal dependence demonstrated in earlier studies (Masumoto and Meyers, 1998; Zhuang et al., 2013; Menezes and Vianna, 2019; Nagura and McPhaden, 2021).

Considering the role of meridional overturning circulation in the mass and heat transport, it is important to examine the year-to-year variability in this circulation. We confirmed in our study that the interannual variability of the meridional transport is

related to the zonal gradient of SLA between the western and eastern regions of the subtropical SIO. In both numerical simulations, the zonal SLA gradients are related to ENSO characterized by positive SLA gradients during El Niño and negative gradients during La Niña conditions. In the ITF-off experiment, the positive SLA gradients associated with the 1997-1998 and 2014-2016 El Niño events are reduced compared to the ITF-on simulation, but generally the SLA gradients

exhibit similar amplitudes and patterns. Nagura and McPhaden (2021) showed that the meridional transport of the subtropical gyre is primarily driven by variability radiated from the eastern boundary. Our analysis shows that local wind forcing modulated by the "atmospheric bridge" effect is an important driver for the meridional transport in the SIO subtropical gyre.

Overall, these findings advance our understanding of regional heat content and sea level variability in this key region. To study the far-reaching impacts of the heat accumulation in the SIO, more research employing ongoing observations as well as ocean and climate models is necessary.

**Data availability**

The altimetry products were provided by the Copernicus Marine Environment Monitoring Service (https://marine.copernicus.eu/www.esrl.noaa.gov/psd/). The IOD index is provided by NOAA/PSL using the HadISST1.1
SST dataset (https://psl.noaa.gov/gcos_). The SAM index is obtained from British Antarctic Survey's website (www.nerc-bas.ac.uk/icd/gjma/). The ECCO LLC270 solution is available for download at https://ecco.jpl.nasa.gov/Version5/Alpha. All data needed to evaluate the conclusions in the paper are present in the paper Additional data related to this paper may be requested from the authors.

**Author contribution**

M.K., D.L.V., K.P., and H.Z. conducted the research and/or aided in the analysis. M.K. led the writing of the manuscript with additional content and/or editorial contributions from all the co-authors. D.L.V. and H.Z. designed the numerical experiments and performed the simulations. All authors approved the final manuscript.

**Competing interests**

The authors declare that they have no conflict of interest.

**Acknowledgements**

M.K. and D.L.V. acknowledge support from NASA Ocean Surface Topography Science Team program (grant NNX17AH59G). M. K., D.L.V., and K.P. were supported in part under the auspices of the Cooperative Institute for Marine and Atmospheric Studies (CIMAS), a Cooperative Institute of the University of Miami and NOAA (cooperative agreement NA20OAR4320472), and/or under a grant from the NOAA Climate Variability Program (GC16-212). M.K., D.L.V. and
K.P. also acknowledge additional support from the NOAA Atlantic Oceanographic and Meteorological Laboratory. H.Z. is





supported by NASA Modeling, Analysis, and Prediction (MAP) and Physical Oceanography (PO) programs. Gratitude is also extended to A. L. Gordon and A. Napitu for sharing the ITF transport data in Makassar Strait, and to R. C. Perez for helpful comments.

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





**Table 1: Summary of the parameters used in the RG model.**

| SLA forcings | Simulation | $c_R$ (cm s$^{-1}$) | g' (m s$^{-2}$) | $\varepsilon^{-1}$ (years) |
|---|---|---|---|---|
| SLA at 13°S and 110°E | ITF-on | 13.0 | 0.07 | 2.5 |
| | ITF-off | 13.0 | 0.07 | 3.3 |
| SLA at 25°S and 110°E | ITF-on | 6.5 | 0.05 | 1.9 |
| | ITF-off | 6.5 | 0.04 | 1.2 |

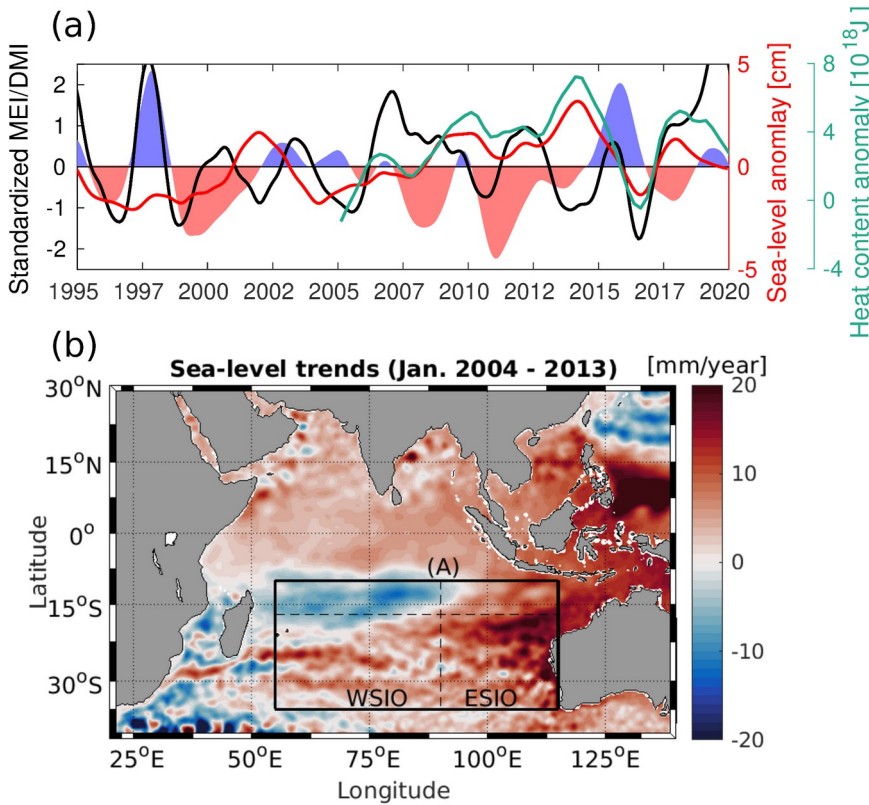


**Figure 1: Climate indices vs. sea level and heat content changes. (a) Time series of the monthly standardized MEI (El Nino/La Nina are shown by blue/red shading) and DMI index (black line). Monthly sea-level anomaly (SLA) from altimetry (red line) and seasonal heat content anomaly from 0 to 2000 m depth (green line) from World Ocean Database (Levitus et al., 2012) are also represented, both averaged over 55°E to 115°E and 10°S to 30°S (box A in fig. b). All the time series are low pass filtered with a cutoff period of 1 year. (b) Dynamic Sea Level trends in 2004-2013 (global mean sea level is subtracted). Boxes in (b) marked the**
**distinction between the tropics *vs.* subtropics and eastern *vs.* western SIO regimes.**





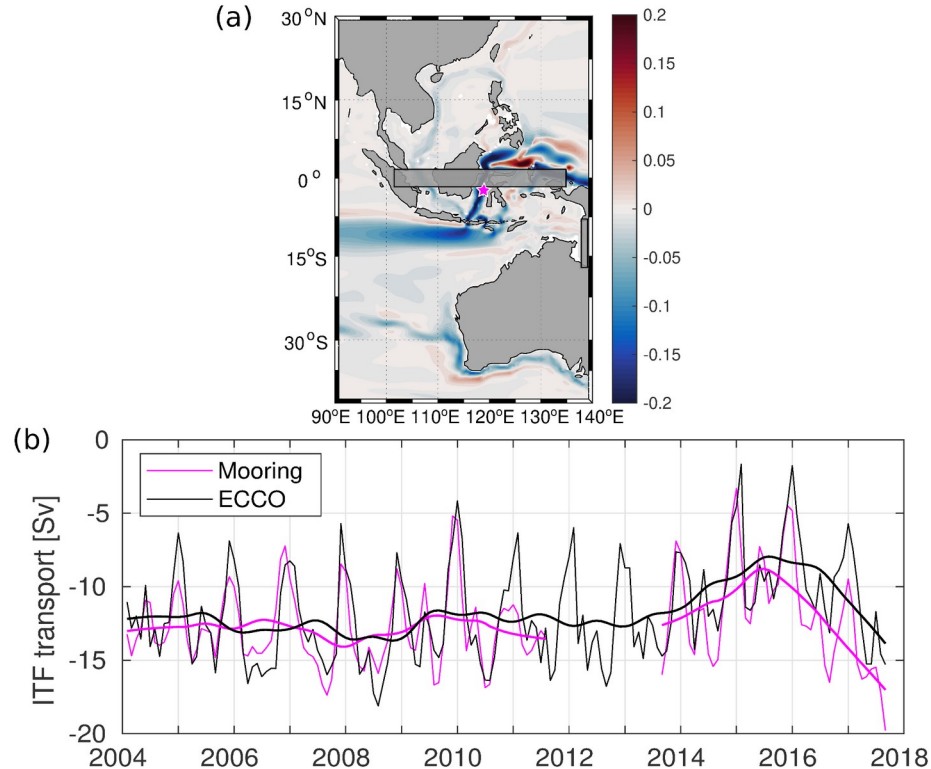

**Figure 2: (a) Difference between the time-mean absolute ocean current velocities (m s$^{-1}$) averaged over the upper 100 m in the ITF-off and the ITF-on numerical simulations. (b) Monthly ITF transport in the upper 760 m from the ITF-on numerical simulation (black line) and in situ velocity measurements (magenta line) from subsurface moorings across the Makassar Strait (magenta star in fig. a). The bold lines represent the low pass filtered time series with a cutoff period of 1 year.**


50




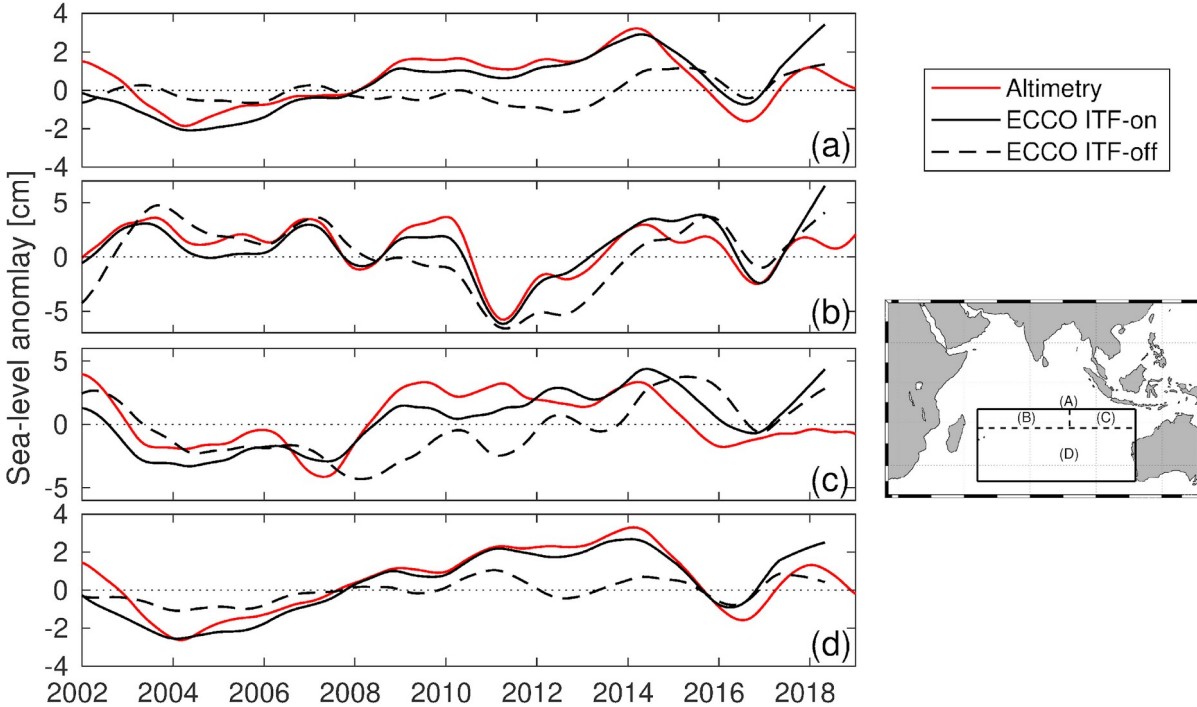

**Figure 3: Time series of the sea-level anomaly (SLA) from altimetry (red line), and from numerical simulations (ITF-on: black line; ITF-off: black dashed line). The SLA (a, b, c, d) are averaged, respectively, in four areas (A, B, C, D) represented in the map on the right panel. All the time series are low pass filtered with a cutoff period of 1 year.**




**Figure 4: Hovmöller diagrams of (a) the observed SLA; (b) the ECCO modeled SLA from the ITF-on simulation; (c) the ECCO modeled SLA from the ITF-off simulation, averaged between 10°S and 30°S. (d) Time series of SLA from altimetry (red line) and from numerical simulations (ITF-on: black line; ITF-off: black dashed line) averaged between 10°S and 30°S at 110°E. All the time series are detrended and low pass filtered (with a cutoff period of 1 year).**




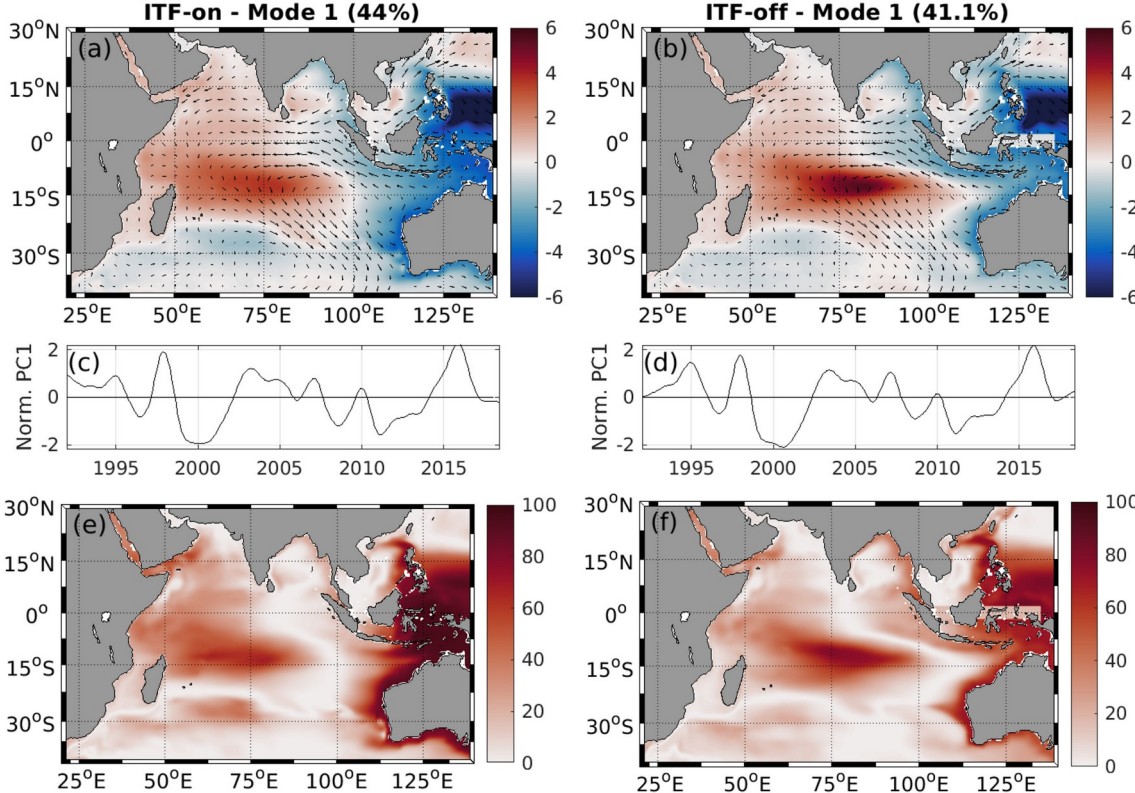

**Figure 5: Regression of monthly SLA (color) and wind stress (arrow) on the standardized principal component of the first EOF mode (PC1) from 1992 to 2018, for the ITF-on (a) and ITF-off (b) simulations, expressed in centimeters per 1 STD change of the respective PC1. (c, d) Time evolution of the associated PC1 normalized by their standard deviations. (e, f) Explained variance between the modeled and the reconstructed SLA.**




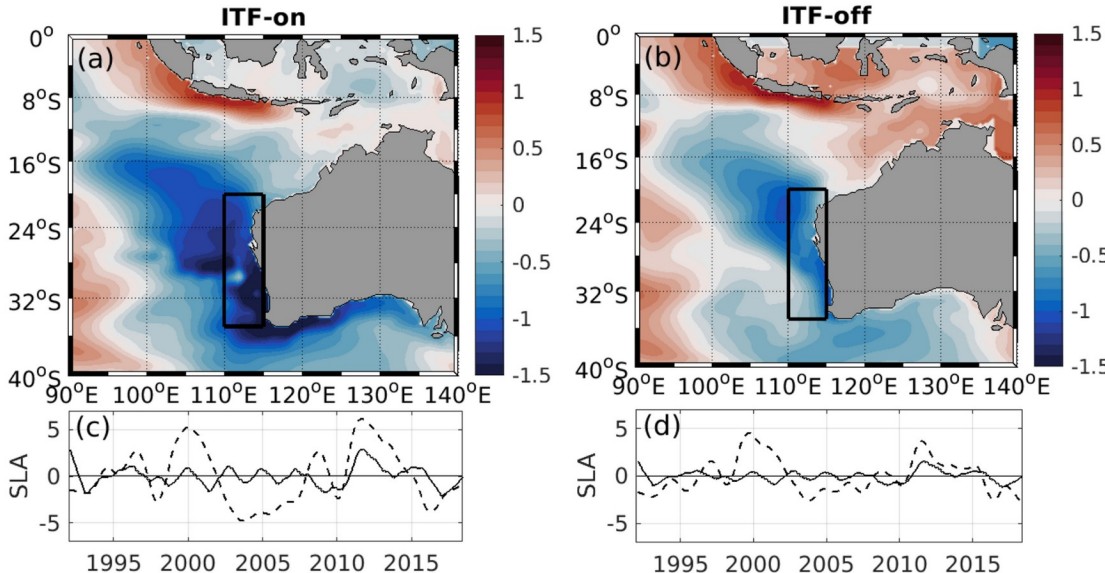

**Figure 6: Regression of monthly SLA on the normalized meridional surface wind stress from 1995 to 2018 averaged in the black box, for the ITF-on (a) and ITF-off (b) simulations, expressed in centimeter per 1 STD change of the meridional surface wind stress (1 STD = 0.045 N m$^{-2}$). (c, d) Time evolution of the reconstructed SLA (black line) and modeled SLA (black dashed line)**
705 **associated with the regression from the ITF-on and ITF-off simulations.**



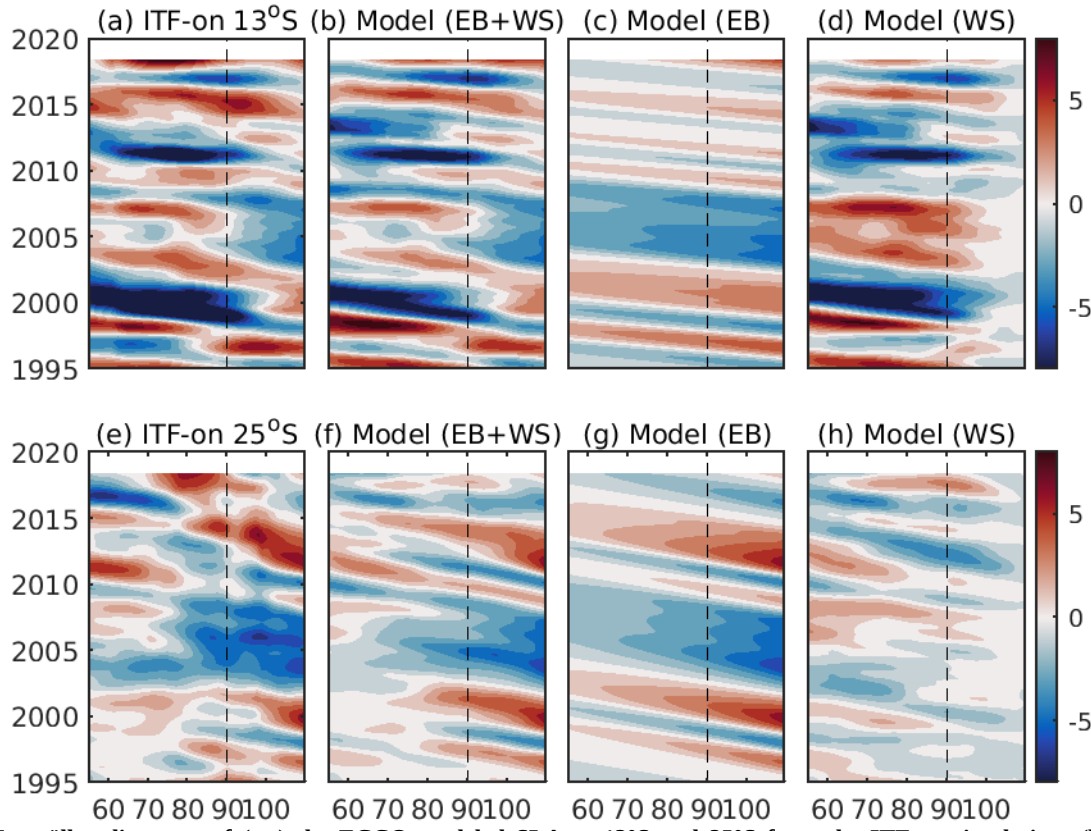

**Figure 7: Hovmöller diagrams of (a,e) the ECCO modeled SLA at 13°S and 25°S from the ITF-on simulation; (b,f) the RG modeled SLA using both the eastern boundary and the local wind stress curl forcing; (c,g) the RG modeled SLA obtained using the eastern boundary forcing only, and (d,h) the RG modeled SLA obtained using the local wind stress curl forcing only. The results from the ITF-on simulation at 13°S are shown in the top panels, and at 25°S in the bottom panels.**

60



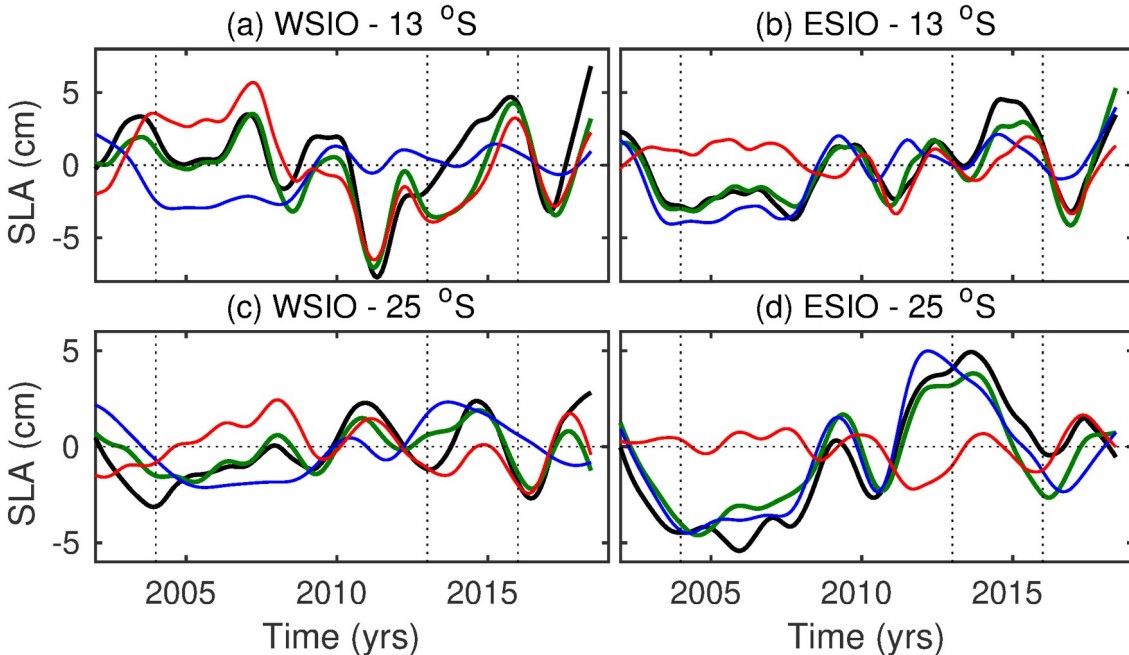

**Figure 8: Time series of the simulated ECCO LLC270 SLA from the ITF-on simulation (black curves) and RG modeled SLA using the eastern boundary forcing (blue curves), the local wind stress curl forcing (red curves), and both forcing terms (green curves), averaged between (a, c) 55°E to 90°E and (b, d) 90°E to 110°E at 13°S (a, b) and 25°S (c, d).**

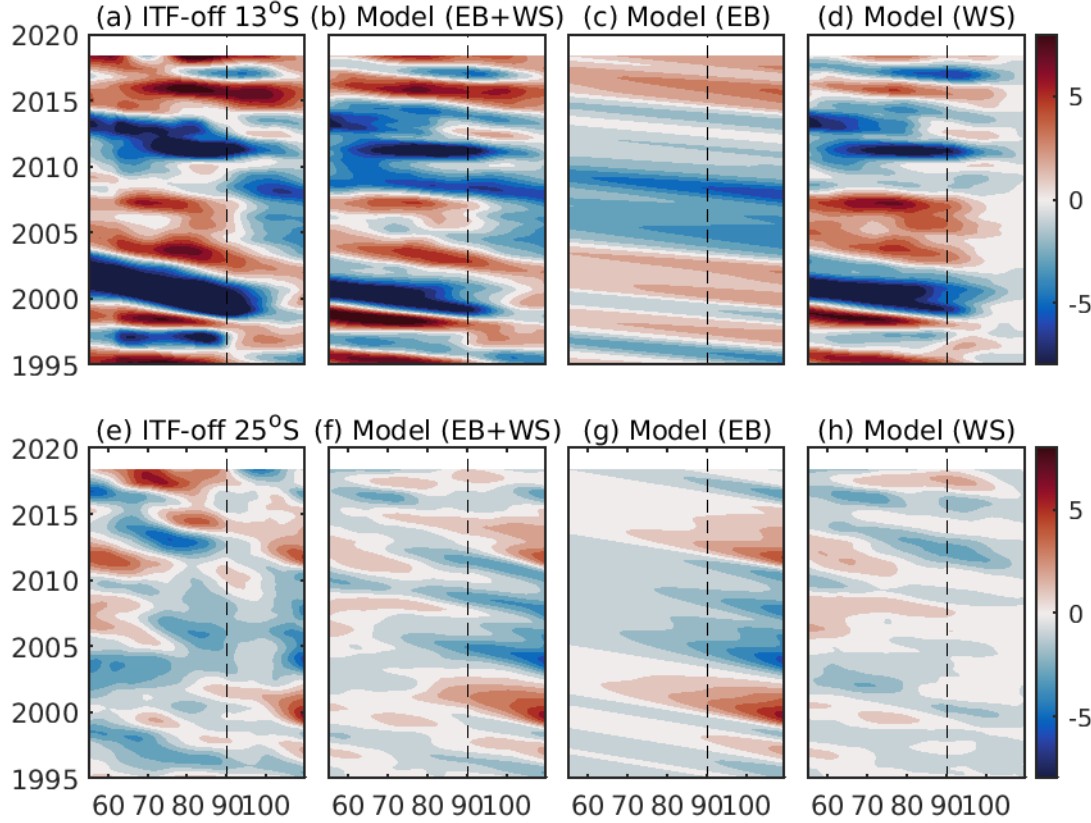

**Figure 9: Hovmöller diagrams of (a,e) the ECCO modeled SLA at 13°S and 25°S from the ITF-off simulation; (b,f) the RG modeled SLA using both the eastern boundary and the local wind stress curl forcing; (c,g) the RG modeled SLA obtained using the eastern boundary forcing only, and (d,h) the RG modeled SLA obtained using the local wind stress curl forcing only. The results from the ITF-off simulation at 13°S are shown in the top panels, and at 25°S in the bottom panels.**

715



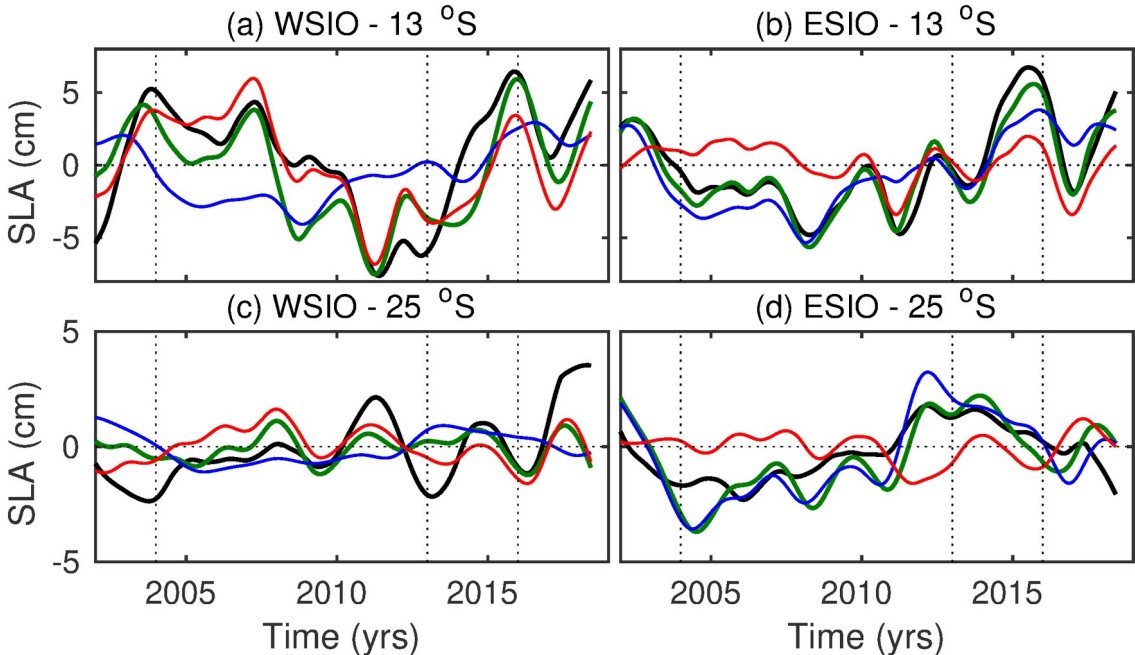

**Figure 10: Time series of the simulated ECCO LLC270 SLA from the ITF-off simulation (black curves) and RG modeled SLA using the eastern boundary forcing (blue curves), the local wind stress curl forcing (red curves), and both forcing terms (green curves), averaged between (a, c) 55°E to 90°E and (b, d) 90°E to 110°E at 13°S (a, b) and 25°S (c, d).**

720





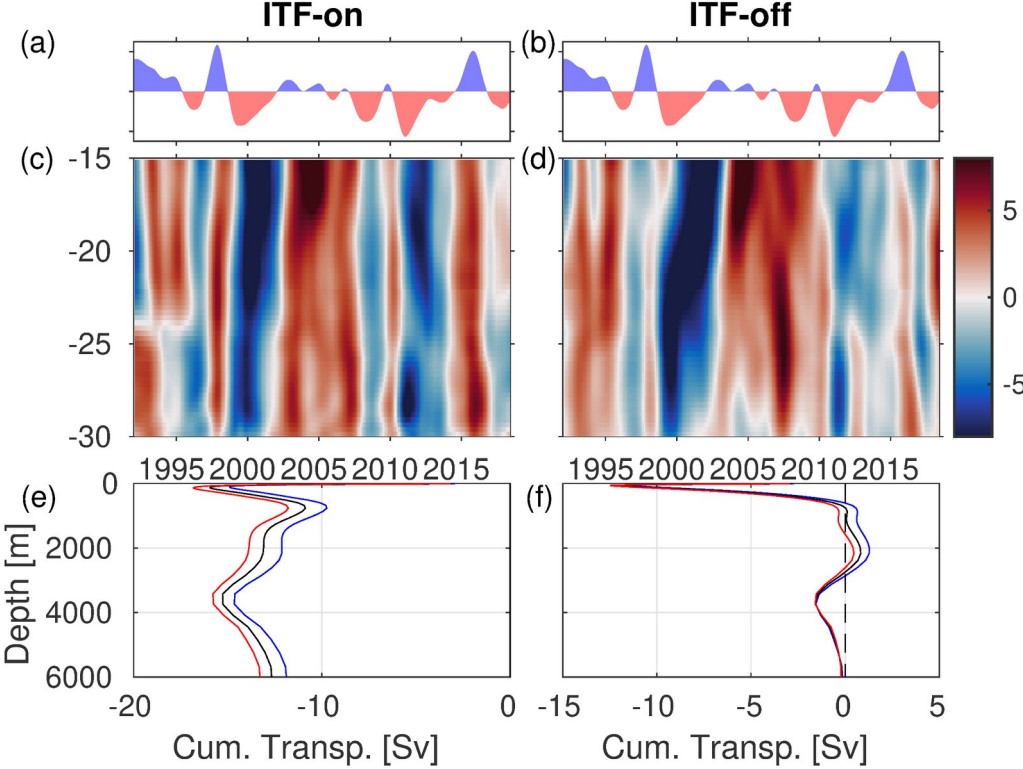

**Figure 11: (a,b) Time series of the standardized MEI (El Nino/La Nina are shown by blue/red shading). (c, d) Zonal differences of SLA (cm) computed between 50°-55°E and 110°-115°E (SLA_west - SLA_east) as a function of latitude and time in the ITF-on and ITF-off experiments. (e,f) Time-mean stream function of the meridional flows across 15°S (black line), and the composite of the stream function associated with El Niño (blue line) and La Niña (red line) in the ITF-on and ITF-off experiments.**