# Peer review of "Interannual variability of sea level in the South Indian Ocean: Local"

_Ocean Science, 2021_

## Author Comment (AC1)

**Response to Reviewer 1**

This study examines the impact of local and remote wind forcing on interannual sea surface height (SSH) variability in the south Indian Ocean, based on numerical experiments with an ocean general circulation model (OGCM). Many past studies worked on this issue using a 1.5layer, reduced gravity long Rossby wave model that possibly suffers adopted approximations. The current study uses an OGCM and has an advantage over past studies. It is potentially worth publication, but I have a question about the setting of numerical experiments. This and other comments are listed below. I recommend major revision.

We would like to thank the reviewer for the constructive and helpful remarks. We have tried addressing them all and we hope that the revised version of the manuscript has been improved. Below we detail the changes we have made to the manuscript, addressing point by point all the issues raised in the review. The comments from the reviewer are shown in bold, while our responses are interspersed between the comments in non-bold text. Changes made to the manuscript are finally listed in italic, whereby page and line numbers indicated in our responses correspond to the new version of the manuscript.

**Major comment**

1) In the ITF-off experiment, the Indonesian archipelago is blocked by land. This eliminates propagation of oceanic waves from the Pacific to the Indian Ocean, but possibly allows wave propagation from the equatorial Indian Ocean to the west Australian coast along the eastern boundary. This route is unrealistic, because waves in the equatorial Indian Ocean intrude into the Indonesian Seas (Durland and Qiu 2003; Syamsudin and Kaneko 2004) and leak out of the basin (Wijffels and Meyers 2004) if the topography is realistic (i.e., there is an opening in the Indonesian archipelago). ENSO excites zonal wind variability in the equatorial Indian Ocean via changes in the Walker circulation (e.g., Xie et al. 2002), which excites equatorial waves (e.g., Chambers et al. 1999; Feng and Meyers 2003). These waves might propagate to the west Australian coast in the ITF-off experiment. Thus, I wonder if ENSO impacts SSH variability along the west Australian coast through zonal wind variability in the equatorial Indian Ocean in the ITF-off experiment, which is an artificial process owing to the experimental setting and does not happen in observations. Please check this possibility and add discussions to the manuscript.

We thank the reviewer for bringing this important and likely issue to our attention. As the reviewer reasonably noted, the closure of the Indonesian throughflow in the ITF-off experiment could have generated an artificial waveguide allowing long waves originating in the equatorial Indian Ocean to propagate all the way to the west Australian coast. In the revised version of the manuscript, we used the daily SSH output to investigate the wave propagation in both the ITF-on and the ITF-off experiments. We established that in both experiments there is a discontinuity in the propagation of coastal trapped waves along the Maritime continent and the northwest-west Australia coasts. Specifically, the waves originating in the equatorial region apparently dissipate in the internal Indonesian seas before reaching the southern coast of Timor island. Coastal trapped waves are also found along the northwest and west Australia coast, but no relation between these waves and the waves along the Maritime continent was found. We have added some discussion about this analysis along with a figure to the revised version of the manuscript.

*l.* 250-276; p. 8-9: "ENSO excites zonal wind variability in the equatorial Indian Ocean via changes in the Walker circulation (e.g., Xie et al. 2002), which generates equatorial Kelvin waves (e.g., Chambers et al. 1999; Feng and Meyers 2003). These waves can eventually get trapped along the Indonesian coast and partly propagate southward. The coastal trapped waves intrude into the Indonesian seas through passages between the islands (Molcard et al., 1996; Durland and Qiu 2003; Syamsudin and Kaneko, 2004; Pujiana et al., 2013; Pujiana and McPhaden, 2020) and then penetrate the Western Pacific (Wijffels and Meyers 2004; Yuan et al., 2018) without significantly affecting the dynamics along the Australian coast. In the ITF-off experiment, an artificial wall was placed to close the Indonesian passages and the Torres Strait. Theoretically, such a wall may permit waves originated in the eastern equatorial Indian Ocean to follow the artificial coastline, pass through the Indonesian Archipelago, and reach the western coast of Australia. If this is the case, then closing the ITF would generate spurious variability in the ESIO.

To verify this possibility, we carried out a cross-correlation analysis of the daily low-pass filtered (with a cutoff period of 1 year) SLA output at a number of locations along Indonesian and Australian coastlines in both the ITF-on and the ITF-off experiment (Figure 3). Apparent signal propagation is observed between Points 1 and 4, but not traceable in the Timor Passage (Point 5) and further east, indicating that the incoming Indian Ocean Kelvin wave energy transmits southeastward along the southern coasts of Sumatra, Java and Nusa Tenggara, with part of the energy making its way into Makassar Strait, Most coastal trapped wave energy appears to leak and dissipate in the internal Indonesian seas. Both experiments exhibit similar cross-correlation functions with maximum values at the same time lags for Points 1-4 (Figure 3 b,c). It takes about 10 days for the signal to propagate from the equator to Point 4 (the distance of approximately 2500 km), which yields the wave phase speed of about 3 m s-1. Further east at Points 5-7, the cross-correlations do not suggest a continued wave propagation (Figure 3b,c). The propagation of coastal trapped waves is also observed along the northwestern and western Australian coast (Figure 3a). The lagged correlations are observed between Points 8 and 12, and the high-frequency SLA variability at these points is uncorrelated with the high-frequency SLA variability in the Torres Strait at Point 7 (Figure 3d,e). In both experiments, it takes about 7 days for a coastal trapped wave to propagate approximately 5000 km from Point 8 to Point 12, which yields the phase speed of about 5 m s-1. Overall, coastal trapped waves appear to propagate and dissipate similarly in both the ITF-on and the ITF-off experiments, meaning that the artificial wall created in the ITF-off experiment is unlikely to generate spurious variability in the ESIO."

*Figure 3: (a)* Bottom topography in the ESIO and the locations (1 to 12) used for cross-correlation analysis. Cross-correlation functions between the daily SLA in (b,d) the ITF-on and (c,e) the ITF-off simulations. The locations 1 and 12 are used as reference points to detect waves propagating along the Maritime continent from 1 to 7 (b, c) and along the west

Australian coast from 7 to 12 (d, e). The red shaded arrows connect the peaks of cross-correlation functions associated with coastal trapped waves.

2) As is stated in the manuscript, winds along the Australian coast explain only 11 or 15% of local SSH variability (Line 311-313 and 446-448). This low ratio can be visually confirmed by the discrepancy between solid and dashed lines in Figs. 6c and 6d. The authors do not discuss what explains the remaining variability. I suggest more analysis should be carried out to specify the cause of SSH variability along the west coast of Australia in the ITF-off experiment. As is mentioned in the previous comment, there can be an unrealistic process in the ITF-off experiment.

We are thankful for this remark, to address this point we have added some discussion about the physical processes potentially causing the remaining variability.

l. 365-367; p.12: "The remaining variability can be explained by instabilities of the LC generating mesoscale eddies in the area and coastal trapped waves originating along the northwest coast of Australia (e.g., Zheng et al., 2018)."

A detailed quantification linking these different forcings with the observed variability would go, however, beyond the scope of the paper. As noted in our response to the previous comment, the possibility of an unrealistic wave propagation from the equatorial region towards the west Australia coast in the ITF-off experiment can be rejected (l. 250-276; p. 8-9).

3) In the current study, SSH variability in the ITF-off experiment is attributed to the effect of the atmospheric bridge. However, the authors do not describe how the atmospheric bridge causes wind variability near the west coast of Australia, but guessed its effect from results of the ITF-off experiment. I suggest that they should conduct additional analysis (such as a correlation analysis between an ENSO index and atmospheric pressure and winds near Australia) and discuss how ENSO impacts surface winds near Australia.

The relationship between ENSO and wind forcing in the SIO was described in Volkov et al. (2020). In the revised version of the manuscript, we also included a regression of wind forcing on ENSO in the eastern SIO.

l.350-354; p.11: "As follows from the regression of wind and Ekman pumping anomalies on MEI (Figure 7a), El Niño events (i.e., positive MEI) are associated with weaker trade winds in the SIO and easterly wind anomalies along the equator. This atmospheric circulation pattern favors a negative (into the ocean) Ekman pumping anomaly along the western Australian coast, resulting in upper-ocean warming. The opposite occurs during La Niña events (i.e., negative MEI)."

Figure 7a: Regression of Ekman pumping (color shading) and wind stress (arrows) on MEI, expressed in meters per month per 1 standard deviation change of the index. Negative (positive) Ekman pumping anomalies associated with the upper-ocean warming (cooling) are shown by red (blue) color.

---

## Author Comment (AC2)

**Response to Reviewer 2**

The authors evaluate in detail the relative importance of local wind forcing and remote oceanic signals from Pacific to the sea level variability in the SIO by using the ECCO simulations and a 1.5-layer reduced-gravity model. The paper is well written and informative. However, there are still some issues that need further clarification. Specific comments are listed as follows.

We thank the reviewer for comments and helpful suggestions. Below we detail the changes we have made to the manuscript, addressing point by point all the issues raised in the review. The comments from the reviewer are in bold, while our responses are interspersed between the comments in non-bold text. Changes made to the manuscript are finally listed in italic, whereby page and line numbers indicated in our responses correspond to the new version of the manuscript.

**1. I agree with reviewer #1 that the closure of ITF passages will create a new wave guide from equatorial Indian Ocean to the west Australian coast, which may lead to overestimation of the eastern boundary forcing in the ITF-off experiment.**

We thank the reviewer for pointing this out. As mentioned in our response to reviewer #1,we have evaluated the possibility of coastal trapped waves to propagate from the equatorial region all the way to the west Australia coast. In the revised manuscript, we have shown that coastal trapped waves propagating along the Maritime continent do not pass the southern coast of Timor island and they are unrelated to the waves found along the northwest-west Australia coast in both the ITF-on and the ITF-off runs. We have added some discussion about this analysis along with a figure to the new version of the manuscript (please refer to the response to reviewer #1 for details).

Furthermore, I have some additional concerns about eastern boundary signals in the ITF-off experiment: (1) Researchers usually adopt free running ocean models to conduct numerical sensitivity experiments. However, as introduced in the section 2.1, the ECCO experiments used in this study are not "free-run" simulations, but constrained by a variety of ocean observations through the adjoint method. This method try to minimize the misfit between observations and simulations by iteratively optimizing the initial conditions, surface atmospheric state and internal parameters. Therefore, the differences between the ITF-on and ITF-off experiments will potentially be reduced by such a data assimilation scheme. As shown in Figure 4, the eastern boundary SLA variability in the ITF-off experiment highly resembles that in the ITF-on experiment, albeit of a relatively weaker amplitude. But these SLA signals are poorly explained by local wind forcing (Figures 6c and 6d). I wonder whether these results reflect the impact of data assimilation scheme applied in both experiments.

It appears that our first version of the paper was not clear on this issue, our apologies. We have changed sentences in the text to clarify that the adjoint method was only used to adjust the model control parameters (Mazloff et al., 2010; Wunsch et al., 2013). So neither the ITF-on nor the ITF-off experiments have data assimilation involved during the integration. They were both free runs using already adjusted parameters.

l. 141-142; p.5: "The ECCO solutions are then obtained by forward unconstrained model integrations using the optimized control parameters."

(2) Is the Torres Strait closed as well in the ITF-off experiment? As mentioned in Lines 227 & 291, the closure of the ITF leads to a weaker LC in the ITF-off experiment. It is known that the LC is a counter-wind flow driven by the poleward pressure gradient. When closing the Indonesian straits, the LC should reverse to an equatorward flowing coastal jet, similar to the Benguela Current in south Atlantic and the Peru Current in south Pacific. Therefore, a southward LC in the ITF-off experiment, despite its weaker speed, may also be induced by the unrealistic adjustment of data assimilation processes in ECCO or by the potential wave transmissions from the Torres Strait. Please check the flowing direction of LC in the ITF-off experiment.

Regarding the mean circulation in the ITF-off experiment, the modeled circulation is very similar to that reported by Lee et al. (2002) in their ocean general circulation model with closed Indonesian passages. Lee et al. (2002) showed that the closure of the ITF induces a weaker SEC and weaker southward flowing LC in the top 100 m. Similar results are obtained in our numerical simulation, the LC is not reversed in the ITF-off experiment (see Figure).

Figure: Time-mean absolute ocean current velocities (m s-1) and direction (arrows) averaged over the upper 100 m in the ITF-off numerical simulation.

Regarding the Torres Strait, yes, the strait is also artificially closed in the ITF-off experiment. Thanks to your previous comment and the one of the other reviewer, we have evaluated the potential wave transmission from the Torres Strait and showed that the highfrequency SLA variability along the western Australian coast is uncorrelated with the high-frequency SLA variability in the Torres Strait at Point 7 (please refer to the response to reviewer #1 for details). We hope that our enhanced discussion of the ECCO solutions (forward unconstrained model) also helps to clear up this point.

**2. Line 8: South Indian Ocean (SIO); Line 26: Southern Indian Ocean (SIO). Please make them consistent.**

As suggested, the definition of SIO was replaced to be consistent through the manuscript.

**3. The ticks of X axis in Figure 1a have equal spacing, but the time intervals between neighboring labels are not uniform.**

Thanks for pointing this out. You are right, the ticks were not correctly labeled. The figure 1a has been updated.

4. Figure 8c: please explain the meaning of the dashed lines. The blue line and the black line show different varying phases during 2014-2016 but the explained variance reaches 61% (Line 336). Please explain how to calculate the explained variances.

As suggested, the meaning of the dashed lines have been added in the caption of the figures. The methodology to calculate explained variance has been added to Section 2.4.

1.229-231; p.8: "The relative contribution of the eastern boundary and the local wind stress forcing terms is represented by the explained variance, which is calculated as the fraction of variance (F) of a variable *x*, explained by another variable *y*:

$$F = 100\% \times \left[1 - \frac{var(x-y)}{var(x)}\right].$$
"

To clarify this point, we decided to not split the time intervals for computing the explained variance. We mentioned now the explained variances for the full time period.

l. 373-384; p. 12: "In the ITF-on experiment, the RG model reproduces the SLA variability reasonably well (Figures 8a,b; Figure 9, green curves). The fractions of SLA variance in the ECCO model explained by SLA reproduced by the RG model at 13°S (at 25°S) are equal to 69% (64%) in the WSIO and 88% (82%) in the ESIO. The relative contributions of the eastern boundary and the local wind stress forcing terms are estimated by computing the fraction of SLA variance in the total RG model (the sum of the eastern boundary and local wind forcing) explained by the individual forcing components. In the RG model, the simulated SLA variability in the ESIO is dominated by the eastern boundary forcing (Figure 8c,g; Figure 9b,d), which explains 62% (90%) of the SLA variance at 13°S (25°S). The local wind forcing becomes the main driver of the SLA variability in the WSIO at 13°S, explaining 70% of the SLA variance (Figure 8b,d; Figure 9a). At 25°S, none of the forcing components explains any variance in the WSIO. Nevertheless, the amplitudes of the two forcing components are similar, meaning that their contribution to the SLA variability is comparable (Figure 8g,h; Figure 9c). Overall, the local wind stress curl over the WSIO is able to either strongly modify the SLA originating from the eastern boundary or generate new anomalies that also propagate westward."

**5. Line 367: "the correlation between the west-east SLA differences" should be "the correlation between the MEI and the west-east SLA differences".**

Thanks for catching that typo. Fixed as suggested (l. 412; p.13).

**6. Line 421: "the observed decade-long heat accumulation is due to the ocean tunnel effect, linked to the ENSO variability." Does it reflect the impact of PDO variability during hiatus period?**

The ITF responds to both ENSO and PDO; a positive ENSO or PDO phase corresponds with a reduced throughflow transport. But the ITF and PDO relationship appeared to be decoupled during the hiatus period for unknown reasons (Li et al., 2018).

**References**

Li, M., Gordon, A. L., Wei, J., Gruenburg, L. K., & Jiang, G. (2018). Multi-decadal timeseries of the Indonesian throughflow. Dynamics of Atmospheres and Oceans, 81, 84-95.

Mazloff, MR, Heimbach, P and Wunsch, C (2010) An eddy-permitting Southern Ocean state estimate. Journal of Physical Oceanography 40(5), 880–899. doi: 10.1175/2009jpo4236.1.

Wunsch, Carl, and Patrick Heimbach. (2013) Dynamically and kinematically consistent global ocean circulation and ice state estimates. In Ocean Circulation and Climate: A 21 Century Perspective, ed. Gerold Siedler, Stephen M. Griffies, John Gould and John A. Church, vol. 103 of International Geophysics, 553–579. Oxford, UK: Academic Press.